# Mobile Colistin Resistance (*mcr*) Genes in Cats and Dogs and Their Zoonotic Transmission Risks

**DOI:** 10.3390/pathogens11060698

**Published:** 2022-06-17

**Authors:** Afaf Hamame, Bernard Davoust, Zineb Cherak, Jean-Marc Rolain, Seydina M. Diene

**Affiliations:** 1Faculté de Pharmacie, IRD, APHM, MEPHI, IHU-Méditerranée Infection, Aix Marseille University, 19-21 Boulevard Jean Moulin, CEDEX 05, 13385 Marseille, France; afafhamame@gmail.com; 2IHU-Méditerranée Infection, 19-21 Boulevard Jean Moulin, CEDEX 05, 13385 Marseille, France; bernard.davoust@gmail.com; 3Faculté des Sciences de la Nature et de la Vie, Université Batna-2, Route de Constantine, Fésdis, Batna 05078, Algeria; cherak-zineb@hotmail.com

**Keywords:** colistin resistance, *mcr* genes, pets, cats, dogs, zoonotic transmissions, colistin usage

## Abstract

**Background**: Pets, especially cats and dogs, represent a great potential for zoonotic transmission, leading to major health problems. The purpose of this systematic review was to present the latest developments concerning colistin resistance through *mcr* genes in pets. The current study also highlights the health risks of the transmission of colistin resistance between pets and humans. **Methods**: We conducted a systematic review on *mcr*-positive bacteria in pets and studies reporting their zoonotic transmission to humans. Bibliographic research queries were performed on the following databases: Google Scholar, PubMed, Scopus, Microsoft Academic, and Web of Science. Articles of interest were selected using the PRISMA guideline principles. **Results**: The analyzed articles from the investigated databases described the presence of *mcr* gene variants in pets including *mcr-1*, *mcr-2*, *mcr-3*, *mcr-4*, *mcr-5*, *mcr-8*, *mcr-9,* and *mcr-10*. Among these articles, four studies reported potential zoonotic transmission of *mcr* genes between pets and humans. The epidemiological analysis revealed that dogs and cats can be colonized by *mcr* genes that are beginning to spread in different countries worldwide. Overall, reported articles on this subject highlight the high risk of zoonotic transmission of colistin resistance genes between pets and their owners. **Conclusions**: This review demonstrated the spread of *mcr* genes in pets and their transmission to humans, indicating the need for further measures to control this significant threat to public health. Therefore, we suggest here some strategies against this threat such as avoiding zoonotic transmission.

## 1. Introduction

Across history, humans have cared for certain animals and continue to share their daily lives with pets, especially cats and dogs. Approximately 223 million pets are owned worldwide today [1]. These animals provide psychological support, and they are faithful and enjoyable companions [2]. However, owners are frequently exposed to pathogenic and zoonotic bacteria harbored by their pets [3,4]. Considerable care is often given to companion animals in order to prevent them from catching infections. For this purpose, several antibiotics including polymyxin are regularly administered as prophylaxis to pets, including those that are also authorized for human use [5]. Antibiotics are widely considered to be the most effective treatment for bacterial infections in pets; hence, their loss of efficacy can result in serious therapeutic difficulties for animals and humans [6]. Colistin (polymyxin E) is a cationic antimicrobial peptide that was first discovered in 1947 and was first used in human medicine in 1959 but abandoned in the 1980s because of its nephrotoxic and neurotoxic effects [7]. However, colistin, which is a powerful molecule, was reintroduced because of the emergence of multidrug-resistant Gram-negative bacteria (GNB), especially those resistant to carbapenems [8]. Polymyxin has been commonly used in veterinary medicine for both pets and food animals for several years. On the other hand, polymyxin E is a last-resort antibiotic against certain multidrug-resistant bacterial infections [7]. Alongside colistin use, colistin-resistant bacteria emerge through two mechanisms: (i) via chromosomal gene mutations such as mutations of the two-component system (TCS) *phoP*/*phoQ* and *mgrB* regulator leading to the modification or the complete loss of the membrane lipopolysaccharide layer, and in addition, several mutations that activate the *pmrA*/*pmrB* (TCS) are characterized as polymyxin resistance mediators [9,10]; (ii) via a mobile colistin resistance gene (*mcr*) on a conjugative plasmid discovered in 2015 by Liu et al. in *Escherichia coli* isolated from pigs in China, thus generating a global health interest [11]. Currently, a total of ten *mcr* variants (*mcr-1* to *mcr-10*) have been reported in the literature [12,13,14,15,16,17,18,19,20]. A few years after the discovery of *mcr* genes, this mechanism has been disseminated worldwide and reported in humans, animals, food, and the environment, with some of them in zoonotic cases [21,22,23,24]. Colistin resistance genes have been evidenced in animals more than in humans [25]. Thus, animals may play a crucial role in the transmission of mobile colistin resistance genes (*mcr*) [26,27]. The close contact between humans and pets increases favorable conditions for the transmission of antibiotic-resistant bacteria through their tight behavior [28]. Previously, several antibiotic resistance genes (ARGs) that were discovered in dogs, cats, and birds were shared with their owners [28,29,30]. Colistin resistance genes (*mcr*) are usually located on mobile genetic elements (i.e., transposons) mainly vehiculated by conjugative plasmids which facilitate the dissemination of *mcr* genes among bacteria [31,32]. Humans and animals can exchange diverse bacterial antibiotic resistance genes (ARGs) via physical contact. The zoonotic transmission of bacteria harboring ARGs is more frequent when it covers food animals or animals in direct contact with humans such as pets [33]. Exchanging plasmids carrying ARGs accentuates the risk of zoonotic transmissions leading to infectious diseases difficult to treat [34,35,36,37,38,39]. In the present review, we report colistin-resistant bacteria in pets (i.e., dogs and cats) and discuss the risk of zoonotic transmission of such bacteria from these animals to their owners.

## 2. Material and Methods

### 2.1. Design and Collection of Articles

This systematic review was conducted according to meta-analysis guidelines (PRISMA) [40]. Harzing’s Publish software, Version 3, Anne-Wil Harzing (Middlesex University, London, UK) was used to improve the search fields and to scan all published studies [41]. The databases used to perform a global bibliography search were as follows: Google Scholar, Web of Science, PubMed, Scopus, and Microsoft Academic. The keywords used in the systematic search were “colistin resistance”, “polymyxin resistance”, “*mcr* genes”, “pets”, “companion animal”, “dog”, “cat”, and “zoonotic transmission”. An additional online research for discussion results was performed using Z-library [42].

### 2.2. Inclusion and Exclusion Criteria

From all the selected articles, the titles and abstracts were principally extracted and analyzed. Articles that were duplicates, were lacking the full text, featured language constraints, or were off-topic were excluded.

### 2.3. Extraction of the Dataset

All articles reporting colistin-resistant bacteria in pets and zoonotic transmission were thoroughly analyzed. The extracted criteria contained all the information about strains harboring *mcr* genes in dogs and cats. The criteria corresponded to (i) the total number of strains screened, (ii) bacterial species, (iii) sequence type (ST), (iv) geographical location where strains were isolated, (v) sample origin, (vi) year of sampling, (vii) plasmids, (viii) zoonotic transmission, (ix) colistin resistance mechanisms, (x) resistance genes, and (xi) associated diseases. The extracted dataset was used to develop a comparative analysis of the emergence of *mcr* genes in cats and dogs to highlight the human risk of zoonotic transmission.

## 3. Results and Discussion

### 3.1. Bibliographic Research

The bibliographic research resulted in a total of 1231 articles from the interrogated databases such as Google Scholar, PubMed, Scopus, Microsoft Academic, and Web of Science. The flowchart shown in Figure 1 presents the process via which studies were selected. The first screening led to the removal of 84% (*n* = 1033) of the total selected studies because they corresponded to duplicated articles (*n* = 198; 16%) or off-topic articles (*n* = 835; 68%). Other studies were included from the Z-library search (*n* = 56). All the selected studies (*n* = 254) were screened for a second time. During the second screening, 41% (*n* = 104) of studies were excluded due to the following reasons: off-topic, language problems, duplicates, uninteresting studies, and studies concerning pet diseases. Among the final 150 selected studies, 12% (*n* = 18) reported bacteria harboring *mcr* genes in dogs, 1% (*n* = 2) reported bacteria harboring *mcr* genes in cats, and 5% (*n* = 7) reported bacteria harboring *mcr* genes in both cats and dogs. Only four studies reported the zoonotic transmission of colistin-resistant bacteria from dogs and cats to their owners. The characteristics of the selected articles were explored using the statistical parameters presented in Table 1.

### 3.2. Colistin Resistance Mechanisms 

#### 3.2.1. Chromosomic Colistin Resistance

Bacteria are exposed to several antimicrobials or environmental stimuli such as colistin, which is a cationic antimicrobial peptide (CAMP) that leads bacteria to develop strategies to protect themselves [43,44]. One of the described mechanisms of resistance to colistin is mutations, which are the first protection means against the alteration of the bacterial membrane [31,45,46]. Figure 2 schematically illustrates the mechanism of colistin resistance induced by chromosomal mutations. Activation of the two-component system (TCS) PmrA/PmrB generates the upregulation of the operons pmrCAB and arnBCADTEF-pmrE (pmrHFIJKLM). The last cascade of activation stimulates the synthesis and transfer of phosphoethanolamine (PetN) and l-Ara4N to the 4′-phosphate group to lipid A in the membrane lipopolysaccharide (LPS). It should be noted that LPS is the primary target of colistin. The addition of cationic groups to the surface of LPS increases its positive charge and decreases the affinity to polymyxin [47,48]. The PhoP/PhoQ TCS can directly activate the PmrA/PmrB TCS via the PmrD regulator [49,50]. However, direct activation of the operon arnBCADTEF can be managed by PhoP independently of pmrD [51]. In addition, the mgrB and micA genes have negative feedback on TCSs (PhoP/PhoQ). This feedback is monitored by inhibition of the kinase activity on PhoQ and/or by stimulating its phosphatase activity. Any mutations in this system, therefore, lead to the overexpression of PhoP/PhoQ that activates the pmrD activator (pmrA activation) and pagL lipid A deacylation [52,53]. Moreover, other colistin resistance mechanisms have been reported such as (i) the phosphorylation of pmrE or Ugd by Etk (tyrosine kinase) leading to alteration of the LPS structure, (ii) the alterations in Kdo (3-deoxy-d-manno-octulosonic acid), residues of LPS which can induce colistin resistance in *E. coli*, or (iii) mutations in mgrR which increase colistin resistance in *E. coli* [54,55].

#### 3.2.2. Plasmid-Mediated Colistin Resistance

Until late 2015, colistin resistance was related to chromosomic mutations without any proven horizontal transfer. Later, Chinese researchers discovered an IncI2-type plasmid carrying a new colistin resistance gene *mcr-1* (for mobile colistin resistance) in *E. coli* strains from pigs [11]. A few months later, researchers began to screen for the presence of these genes in various samples from different countries around the world [56]. The recent discovery of *mcr-**1* does not exclude its prior existence, as proven by its detection in *E. coli* in a collection dating from the 1980s [57]. Researchers then began to look for genes similar to *mcr-1* which were classified as *mcr* variants ranging from *mcr-2* to *mcr-10* (*mcr-10* is the most recently discovered gene) [20,58]. The *mcr* genes encode phosphoethanolamine transferase enzymes that decrease the affinity of colistin to the bacterial external membrane through the addition of a phosphoethanolamine moiety to lipid A [59]. The colistin resistance mechanism usually confers a low level of colistin resistance with low minimum inhibitory concentrations [60]. Colistin resistance is of great interest due to its ability to be transferred via horizontal gene transfer mechanisms [61]. The significant spread of *mcr* genes is due to the association of *mcr* genes with a large variety of mobile genetic elements [62,63,64]. It is also possible for colistin resistance genes to coexist in the same bacterial isolate via different mobile genetic elements as recently described in the co-occurrence of *mcr* genes in *E. coli* harboring the *mcr-3* gene on an IncFII plasmid and a transposon carrying the *mcr-1* gene in the chromosome [65].

### 3.3. Antimicrobial Uses

#### 3.3.1. Antimicrobials and Colistin Treatment for Pets

In this section, we report colistin use in companion animals around the world. In Europe, 62% of cats with urinary pathologies and 36% of dogs with dental disease are treated with critically important antibiotics (CIAs) including colistin [66]. Colistin use in animals is for therapeutic purposes, as a feed additive, or as metaphylactic treatment [67]. Less stringent standards are imposed upon the use of colistin in companion animals compared to those imposed on humans and food-producing animals [68]. The use of antimicrobials in Europe was investigated in animals, including cats and dogs, indicating that cats receive antimicrobial treatment (13%) less frequently than dogs (25%) [69]. The most commonly used antimicrobials were β-lactams such as amoxicillin–clavulanate (cats: 28%; dogs: 27%). Concerning cats, the second most widely used active compound as an antibiotic was cefovecin (third-generation cephalosporin) [69]. Currently, polymyxin B sulfate represents 6% of the treatments used in companion animals. Moreover, polymyxin B sulfate is classified by the World Health Organization (WHO) as a critically important antimicrobial of the highest priority [69]. Further German studies identified the total amount of polymyxin used in dogs and cats in 2017 and 2018 in one veterinary hospital to be 0.06% and 0.13%, respectively [70]. In recent research carried out in 2019 in Australia on a cohort of cats and dogs using the VetCompass software, the most frequently used antibiotics were cefovecin and amoxicillin–clavulanate, respectively [71]. In 2014, very restricted use of antibiotics in pets (1% of 1.4 million kg of antibiotics sold) was reported in Canada [72]. In Japan, the extent of antibiotics used in domestic animals remains unknown since human antimicrobials are used for companion animals rather than humans [73]. In total, an estimated 29.9 tons of antimicrobial drugs were used in companion animal clinics between 2017 and 2018. Less than 1% of this total represents the annual consumption of polypeptides (colistin) [74].

#### 3.3.2. Antimicrobial Use and Antimicrobial Resistance

Many studies have reported that the increase in AMR results from uncontrolled AMU [75]. The quantitative use and the qualitative use of antibiotics in pets are very significant in the veterinary field. AMU is based on information collected from veterinary centers and pharmaceutical companies. Some studies are not representative due to the lack of information regarding the quantification of AMU [76]. The most commonly used antibiotics in animal medicine are also used in human medicine. Furthermore, the use of some antibiotics is restricted due to their unfavorable and toxic side effects (glycopeptides and streptogramins) [77]. Due to the rapid spread of AMR and the excessive use of antibiotics in pets, restrictions on antimicrobial use were required. Such restrictions were applied to antibiotics that are critical to human health, such as colistin [78]. Colistin was used for more than 70 years before being banned due to its high nephrotoxicity [38]. Although colistin use was prohibited for several years, its resistance continues to emerge. The spread of multidrug-resistant bacteria and especially carbapenem-resistant bacteria has led to the reuse of colistin [68]. Furthermore, in some cases, there are no convenient alternatives to colistin, which is an effective antibiotic. Few drugs are available against certain urogenital infections and respiratory infections in pets [79].

#### 3.3.3. Approaches Aimed at Reducing Antimicrobial Use

Several attempts have been developed in South Africa to optimize antimicrobial use (AMU) and reduce antimicrobial resistance (AMR) in domestic animals. These approaches were based on questionnaires about current knowledge and attitudes toward antibiotics and strategies to restrict their use. According to the survey results, the majority of respondents (79.4%; *n* = 81) believed that antibiotics were sometimes prescribed for unconfirmed infections; on the other hand, 75.5% of respondents (*n* = 77) rarely took antibiotics to treat infections, while 18.7% of respondents (*n* = 19) often used antibiotics for treatment [80]. Other therapeutic approaches have also been used, including strategies to combine colistin with other antibiotics. Antibiotics are combined to increase antibacterial efficacy and reduce the emergence of AMR [81]. Recent extensive research concerning AMU in veterinary medicine has been greatly extended. AMR is highly increasing in human medicine, leading to doubt about all antimicrobial usage, particularly in food animals in which these antibiotics have long been used in centralized management to promote growth and prevent diseases [82]. AMU in dogs and cats has also been investigated in three European countries (Belgium, Italy, and the Netherlands), and no association between AMU and AMR was found in the investigated samples [69]. However, other researchers have reported a direct crosslink between the use of colistin in the European Union and the development of resistance in animals [83]. The principal causes of AMR in companion animals were purported to be related to the quality of use, rather than the quantity [50]. Many aspects of reducing AMU in animals and infection control can be improved in order to reduce the spread of AMR and to preserve antibiotics for future use [68].

### 3.4. mcr Genes in Companion Animals

Among the analyzed articles here, 18 of them reported the detection of *mcr* genes in dogs, while 2 studies reported their detection in cats. Seven studies described the detection of *mcr* genes in both dogs and cats.

#### 3.4.1. *mcr* Genes in Dogs

As presented in Table 2, *mcr* genes were found in dogs in several countries around the world from diverse dog samples. For example, in 2016, one study reported the detection of transferable plasmids harboring *mcr* genes in dogs. This concerned the isolation of four *E. coli* strains belonging to the sequence type ST354 and harboring the *mcr-1* gene on a conjugative replicon. The study was conducted on 39 fecal samples taken from a pet shop in China. Interestingly, one of the isolates carried IMP-4 carbapenemase [84]. Thereafter, three Chinese studies published in 2017 reported the isolation of *mcr-1*-positive *E. coli* and *Klebsiella pneumoniae* from fecal samples, nasal samples, and rectal swabs [85,86,87]. Fecal samples were sampled from dogs living in pig and poultry farms. The swine farm isolates belonged to ST10 and harbored the *mcr-1* gene on an IncX4 plasmid [86]. In 2018, Wang and his colleagues reported the isolation of seven *mcr-1*-producing *E. coli* from clinical samples (urine, nasal secretions, feces, and diarrhea); the strains belonged to ST93, ST1011, ST3285, and a new strain [35]. In 2019, an *mcr-1*-positive *E. coli* ST770 isolate was obtained from a urinary tract infection in Argentina. The *mcr-1* gene was harbored by the conjugative IncI2 plasmid, and the strain co-expressed the *bla*_CTX-M-2_ extended-spectrum β-lactamase (ESBL) [88]. In addition, two Chinese studies reported the isolation of *mcr*-producing *E. coli* ST6316, ST405, ST46, and ST162 from clinical samples [89,90]. In Ecuador, Ortega-Paredes and his colleagues described the isolation of an *mcr-1*-positive *E. coli* strain co-expressing *bla*_CTX-M-65_ ESBL from disposed feces in a public park [91]. Furthermore, *mcr-1* was detected in *K. pneumoniae* ST307 and *Enterobacter cloacae* ST1005 from clinical samples [92]. In 2020, a Brazilian study described the isolation of *mcr-1*-positive *E. coli*, *Enterobacter* sp., and *Klebsiella* sp. from clinical samples [93]. Furthermore, *mcr-1* and *mcr-3.7* were detected in a single *E. coli* ST132 isolated from a fecal sample in China. The *mcr-1* and *mcr-3.7* genes were located on two different transferable (together and separately) plasmids, namely, IncX4 and IncP2, respectively, and the isolate co-produced *bla*_CTX-M-14_ ESBL [94]. In addition, two studies from Ecuador reported the isolation of *mcr-1*-producing *E. coli* ST1630 and ST2170 (from rectal swabs) and ST162, ST1196, and ST744 (fecal samples). The rectal swab isolates harbored the gene on a transferable IncI2 plasmid [95], and fecal isolates co-expressed *bla*_CTX-M-55_ and *bla*_CTX-M-65_ ESBLs [96]. In South Korea, *mcr-1*-positive *E. coli* ST162 was isolated from diarrhea. Like the previous study, the *mcr-1* gene was located on an IncI2 transferable plasmid [97]. In 2021, we noted the detection of other *mcr* genes in samples from dogs. Among seven studies reporting the detection of *mcr* genes in dogs, only three studies from China reported detection of the *mcr-1* gene. This was in relation to the detection of several strains of *mcr-1*-producing *K. pneumoniae* and *E. coli* from fecal samples, rectal swabs, and clinical samples [98,99,100]. The *mcr-1* gene was harbored by the IncI2, IncX4, and IncHI2 plasmids [98]. Except for *mcr-6* and *mcr-7*, all the other *mcr* genes have been reported from dogs. Wang et al. (2021) reported the detection of *mcr-2*, *mcr-3*, *mcr-4*, *mcr-5*, *mcr-9*, and *mcr-10* in *K. pneumoniae* isolates from fecal samples in China. Interestingly, associations between *mcr-1* and *mcr-3* or *mcr-5* were reported [100]. The *mcr-3* gene was also reported in *E. coli* ST10 co-expressing *bla*_CTX-M-55_ isolated from clinical samples in Thailand [101]. The *mcr-8* gene was detected in *K. pneumoniae* ST3410 co-harboring the *bla*_CTX-M_ ESBL gene obtained from a nasal swab in China [99]. The *mcr-9* gene was also detected in Egypt and the United Kingdom. An Egyptian study reported the detection of *mcr-9*-positive *Enterobacter hormaechei* ST493 isolated from clinical samples [102]. A British study described the isolation of *mcr-9*-positive *E. coli* ST372 co-expressing the *bla*_CTX-M-9_ ESBL obtained from clinical samples [103]. Lastly, *Enterobacter roggenkampii* positive for the *mcr-10* gene on an IncFIB plasmid was isolated in Japan from a pus sample [104]. In France, although the use of colistin in dogs is highly monitored, *mcr-1* genes were detected in dogs in our recent work [105].

#### 3.4.2. *mcr* Genes in Cats

Compared with dogs, few data are available regarding the detection of the *mcr* gene in cats (Table 3). A total of six studies reported the isolation of *mcr*-producers from cats. The first one was from China in 2016, where the authors described the detection of *mcr-1*-positive *E. coli* belonging to ST93 and another previously undescribed strain in fecal samples [84]. The second study reported the detection of the *mcr-1* gene in *E. coli* isolated from nasal and rectal swabs from cats in China in 2017 [87]. The third study was also conducted in China and reported the detection of *mcr-1 E. coli* ST93 isolated from a diarrheic cat in 2018 [35]. The fourth study was from Brazil and reported on the isolation of *mcr-1*-positive *K. pneumoniae* ST307 causing urinary tract infection in 2021 [106]. The other *mcr* gene detected in cats was *mcr-*9 in *Enterobacter hormaechei* ST493 and ST182 and *Enterobacter asburiae* from clinical samples in Egypt and from nasal swabs in Japan, respectively [102,107]. In the Japanese isolates, the *mcr-9* gene was located on the IncHI2 plasmid [107].

In addition to the isolation of *mcr*-producers, the direct detection of *mcr* genes in fecal samples from dogs and cats was also reported in China in 2017 and more recently in 2022 [108,109]. Another recent study in France showed that cats host bacteria harboring the colistin resistance *mcr-1* gene [105].

#### 3.4.3. Source of *mcr* Genes in Pets 

Colistin-resistant Enterobacteriaceae in dogs are managed by diverse epidemiological factors [110]. Flies with several resistance genes against cephalosporins (*bla*) and colistin can be a contamination source of *mcr* genes between dogs and their owners [111]. As reported, humans may not be the origin of *mcr* genes in domestic animals. In Beijing, characteristics of *K. pneumoniae* in humans and companion animals were largely different, and *mcr* genes and *bla*_NDM_ were not present in the genomes of *K. pneumoniae* isolated from humans [99]. In another study, researchers suspected that farm animals were the source of *mcr* genes in diseased dogs and cats (without colistin treatment) [100]. The environment is also an important factor in the spread of *mcr* genes such as *mcr*-*1*, *mcr-3,* and *mcr-7.1*, as well as other ARGs [112]. Many methods were performed to review the spread and evolution of ARG transmission from food animals, humans, and the environment [113]. Furthermore, as reported in China, pet food was also identified as a source of intestinal colistin resistance genes in pets [87]. In addition, in Vietnam and Japan, wastewater is a source of *mcr-1* in urban sewage. The health of city dwellers and pets is generally reflected in domestic sewage [114]. The spread of mobile genetic elements is responsible for the high emergence of *mcr* genes across countries and continents. As a result, colistin resistance genes can easily be transferred from humans to animals [115]. To date, mobile genetic elements have played a crucial role in the dissemination of *mcr* genes around the world [32,116]. As shown in Figure 3, the *mcr* genes in dogs and cats have begun to spread around the world.

### 3.5. Zoonotic Transmission

#### 3.5.1. Zoonotic Transmission of *mcr* Genes between Pets and Humans

The zoonotic transmission of *mcr*-carrying enterobacteria was previously suggested in four studies from China and Ecuador. Zhang et al. (2016) suggested a potential transmission of *mcr-1*-positive *E. coli* between dogs and humans. This suggestion was based on the isolation of clonally related *mcr-1-*producing strains (by multilocus sequence typing and pulsed-field gel electrophoresis profiles (PFGE)) from dogs and a pet shop worker in China [84]. Following this study, Lei and his colleagues reinforced this suggestion when they investigated the prevalence of *mcr*-producers in companion animals in Beijing. They observed clonal relatedness (the same PFGE patterns) between *mcr-1*-positive *E. coli* isolated from dogs, cats, and one pet owner. In the same study, pet food samples were positive for the *mcr-1* gene, suggesting that several factors may contribute to the emergence of *mcr-1* in pets [87]. In addition, another Chinese study reported significant *mcr-1* carriage between dogs and their owners, where the carriage of this gene by owners was a risk factor for its presence in dogs [98]. Lastly, the owner of a dog who was tested positive for intestinal carriage of *mcr-1*-producing *E. coli* was reported as the first case of *mcr-1* carriage in Ecuador [95]. Furthermore, pets harbor bacteria that require the use of polymyxin for treatment, including methicillin-resistant *Staphylococcus aureus*, methicillin-resistant *Staphylococcus* spp., vancomycin-resistant *Enterococcus*, and ESBL- or carbapenemase-producing Enterobacteriaceae and Gram-negative bacteria [117]. The *mcr-1* and *mcr-2* colistin resistance genes, in particular, have a high potential for zoonotic transmission (being predominant in animals rather than in humans) [57]. According to one study on 229 Chinese families, there was a significant co-occurrence of *mcr* and β-lactamase genes in dogs and their owners. The *mcr-1* and *bla*_CTX-M_ genes were found to be present in 2.7% and 5.3% of the population, respectively [98]. In Egypt, researchers reported a high potential of animal–human transmission of *bla*_VIM-4_-, *bla*_OXA-244_-, and *mcr-9*-producing *E. hormaechei*. The transmission was associated with respiratory infections. These bacteria also harbored β-lactam and carbapenem resistance genes; this was the first report to confirm their potential animal-to-human transmission [102]. According to another study conducted in Argentina, dogs can carry *mcr* genes and *bla*_CTX-M-2_ co-producing *E. coli* (ST770) previously reported in humans [88]. Due to the close relationship between pets and their owners, the microbiota of pets and humans share a diversity of bacteria and ARGs [118]. Dogs and cats host zoonotic microorganisms due to their close physical contact with humans (licking, petting, and contact with furnishings) including around pets’ spaces (carpets and beds) [119]. The exchange of pathogenic microorganisms and resistance genes between animals and their owners has frequently been reported [120,121]. In some cases, ARGs found in hospital patients were similar to those found in pets. As a result, vigilance is required against the zoonotic transmission of resistant bacteria, which are shared between pets and their owners [122].

#### 3.5.2. Health Risks Associated with Colistin-Resistant Bacteria in Pets

People who bring pets into their homes are the most exposed to the emergence of pathogenic organisms. These individuals are exposed to bacteria that carry the *mcr* gene or bacteria that are naturally colistin-resistant, especially those linked to human diseases. Vancomycin-resistant enterococci (VREs) are naturally colistin-resistant bacteria, and the zoonotic transmission of VREs from pets to their owners is a public health problem [123]. In Guangzhou, China, and South Korea, researchers have reported diarrhea in dogs caused by *E. coli* with *mcr* genes (ST93, ST3285, and ST160) [35,97]. Urinary tract infections have also been diagnosed featuring colistin-resistant bacteria in pets from Europe (France, Spain, and Portugal) and Argentina. Bacterial culture revealed the presence of the multidrug-resistant bacteria *Acinetobacter* spp. and *E. coli* carrying the *mcr-1* gene associated with other genes, namely, *bla*_CTX-M-2_, *aadA1,* and *sul1* [88,124,125]. In China, the UK, and Egypt, pneumonia and respiratory diseases have been diagnosed in canines and their owners. The pneumopathy was associated with the presence of the following colistin-resistant strains: *K. pneumoniae*, *E. coli*, *P. aeruginosa*, and *E. hormaechei* carrying *mcr-1* or *mcr-9* [35,102,126]. Further enteropathogenic and gastrointestinal diseases revealed the presence of *E. coli* (ST372) harboring *mcr-1* isolated from dogs and humans in Spain [108,124]. Other colistin-resistant bacteria linked to infections or nosocomial diseases have also been found in Spain, the United Kingdom, Lebanon, Taiwan, and China [84,92,98,126,127]. The co-occurrence of *mcr* genes with other resistance genes increases the potential of the bacterium to exhibit multiple resistance to antibiotics, which is a major public health problem. Colistin-resistant isolates described in humans and their pets can be associated with β-lactam resistance genes. Genome analysis revealed *mcr* genes associated with these β-lactamases, namely, *bla*_TEM_, *bla*_CTX-M_, and *bla*_SHV_ [87,128,129,130,131,132,133]. Furthermore, carbapenem resistance genes can be adequately supplied with *mcr* genes produced in clinical bacteria such as *bla*_NDM-1_ [134]. The coexistence of *mcr* and carbapenemases indicates the crossover of resistance between pets and their owners. Multidrug-resistant bacteria with *EptA* and β-lactamase genes are considered to be the strains for which it is complicated to find an effective therapeutic solution [128,135].

#### 3.5.3. Strategies to Control the Zoonotic Transmission of Colistin Resistance from Pets

Zoonotic transmission is managed by different bacterial species that can harbor a variety of ARGs through recombinant plasmids. Systematically, this increases the therapeutic difficulty to treat infectious diseases caused by these bacteria [136,137]. The significant spread and horizontal transmission of colistin resistance highlight the need to adopt restriction protocols on colistin use using the One Health approach [138]. Mobile genetic elements (MGEs) play a major role in zoonotic transmission [139]. Figure 4 shows a graphic representation of zoonotic transmission and MGEs that monitor the spread of *mcr* genes.

In this section of the review, we suggest some approaches to reduce zoonotic transmission. Reducing the use of all antibiotics, particularly colistin, would reduce the emergence of colistin resistance. Colistin resistance is usually mediated by chromosomic mutations, *mcr* genes, or the selection pressure of bacteria resistant to colistin [140,141]. Livestock is the first source of ARGs for human beings due to the excessive use of antibiotics as a growth factor or in the form of subtherapeutic treatment. Antibiotic concentrations, on the other hand, should be investigated in relation to the spread of ARGs in order to assess the risk of zoonotic transmission from pets [141]. The lack of daily hygiene and cleanliness is one of the major reasons for the emergence of pathogens and resistance genes [142,143,144]. In order to reduce antibiotic use and gastrointestinal infections, strict hygiene conditions must be observed to ensure uncontaminated food and clean drinking water. Furthermore, global access to vaccination can also play an important role in protecting pets from bacterial and viral infections that require antibiotics and the use of colistin [145]. Pets such as cats and dogs are in direct contact with human beings, and immediate vigilance is required regarding the pet–human mode of transmission [146]. Considering the fact that pets are in close physical contact with humans and frequently share their living quarters, the use of cleaning and disinfection tools is necessary [142,147]. The evacuation of waste from pharmaceutical and manufacturing plants, pet shops, and hospitals must be rigorously monitored. The waste ends up in waterways and can contribute to the emergence of ARGs such as *mcr* genes in soil and water [148]. To reduce the transmission of zoonotic organisms that are pathogenic in Germany, researchers investigated the potential collection of samples from pets and their owners. The current method has proven to be a very useful tool for investigating zoonoses in population-based studies [149]. Moreover, AMR and AMU in pets must be translated into an accessible shared public data system [150].

## 4. Conclusions

The number of adopted pets by humans has increased, and due to the close contact between these pets and their owners, more attention is being given to their welfare. This review confirms the dissemination of plasmid-mediated colistin resistance in this kind of companion animal, where 8 out of the 10 currently described *mcr* genes have been detected in samples obtained from these animals. This concerning resistance mechanism has been detected in both diseased and healthy animals, thus presenting an important reservoir for these genes. Considering that some studies have highlighted the potential zoonotic transmission of *mcr*-producers between pets and their owners, greater emphasis should be given to screening and reducing colistin resistance, especially regarding *mcr* genes in companion animals. Certain behaviors can be adopted to avoid zoonotic transmission. The most important is daily hygiene at home or outside during any contact with pets. The zoonotic transmission of *mcr* genes should be further investigated in order to minimize the consequences of colistin resistance for human health.

## Figures and Tables

**Figure 1 pathogens-11-00698-f001:**
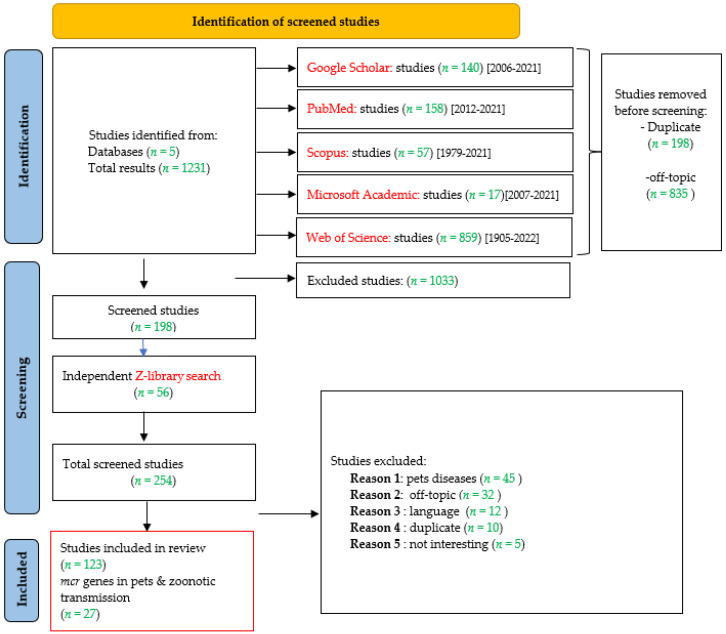
Identification of included studies in this review using PRISMA guidelines (meta-analysis).

**Figure 2 pathogens-11-00698-f002:**
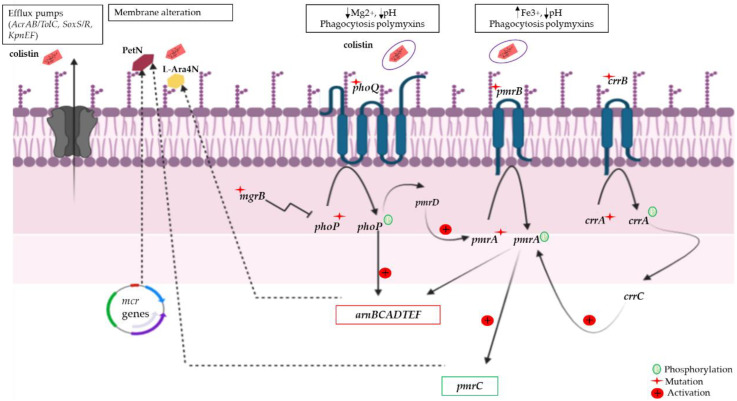
Signaling regulation involved in the colistin resistance mechanism. The gene mgrB exerts negative feedback on the two-component system (TCS) phoP/phoQ. A mutation of the mgrB gene (indicated by red-colored star symbols) induces a constitutive induction of the phoP/phoQ system. Activation of phoP/phoQ activates pmrD and the arnBCADTEF operon; pmrD in turn activates pmrA. The pmrA/pmrB TCS can also be activated by a mutation in the pmrA/pmrB genes; this activation activates both arnBCADTED and pmrC, which collectively modify lipopolysaccharides (LPSs) via the addition of 4-amino-deoxy-l-arabinose (L-Ara4N) or phosphoethanolamine (PetN). PetN can also be added to LPSs by phosphoethanolamine transferase expressed by the mcr genes. Amino acid substitutions in CrrB/CrrA induce crrC expression by inducing elevated expression of pmrC via the activation of pmrA. On the other hand, the efflux pumps (AcrAB/TolC, SoxS/R, and KpnEF) allow rejecting colistin outside the bacteria.

**Figure 3 pathogens-11-00698-f003:**
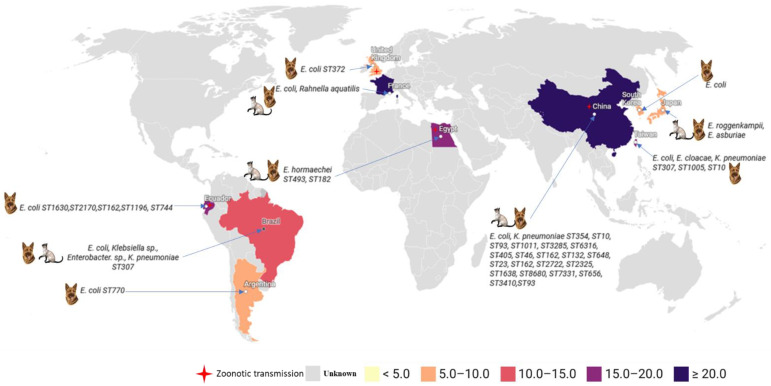
Global distribution of bacteria harboring *mcr* genes isolated from dogs, cats, and zoonotic transmission around the world.

**Figure 4 pathogens-11-00698-f004:**
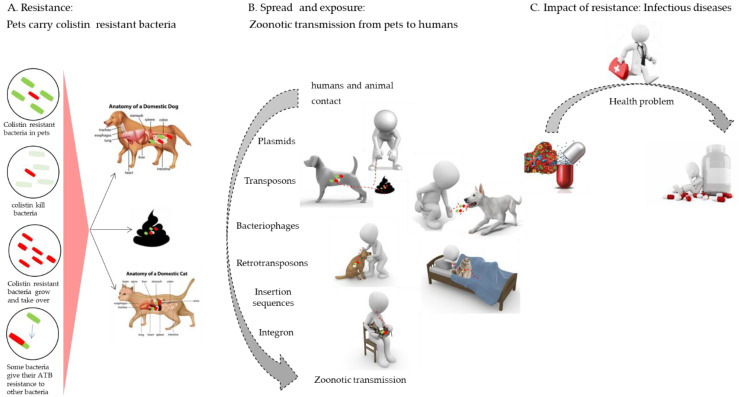
Transfer of colistin resistance from pets to their owners. (**A**): Animals carry in their guts various bacteria in which colistin resistant bacteria have a high prevalence. Those bacteria are found in stools of pets. (**B**): The close contact between pets and their owners exposes human to the zoonotic transmission of colistin resistance via MGEs. (**C**): Zoonotic transmission of MDR bacteria is a public health problem. Small red lines represent the transfer of colistin-resistant bacteria via mobile genetic elements. Red and green bacilli represent bacteria which are resistant and sensitive to colistin, respectively.

**Table 1 pathogens-11-00698-t001:** Characteristics and limitations of the citation metrics and underlying sources that Publish or Perish used for all reporting items obtained during the literature search. The keywords used for the research were as follows: polymyxin resistance, colistin resistance, pets, companion animal, dog, cat, *mcr* genes, and zoonotic transmission.

Source	Papers	H Index	G Index	AWCR	E Index	H Coverage	G Coverage	Year First	Year Last
Google Scholar	140	30	71	1217	60.87	91.2	99.9	2006	2021
Web of Science	859	39	67	2319	46.97	36.73	53.34	1905	2022
PubMed	158	10	9.29	15	9.56	9.76	9.56	2012	2022
Scopus	57	16	23	213	14.04	71.3	83.6	1979	2022
Microsoft Academic	17	4	8	41	6.86	79.7	96.2	2007	2022

**h_index**: quantification of an individual’s scientific research output; **g-index**: the (unique) largest number of the top g articles received (together) with at least **g^2^** citations; **AWCR**: the number of citations of an entire body of work, adjusted for the age of the paper; **e-index:** the square root of the surplus citations in the h-set; **h_coverage** and **g_coverage:** coverage data for citations.

**Table 2 pathogens-11-00698-t002:** The *mcr*-positive isolates detected in dogs.

*mcr* Genes	Bacterial Species	Sequence Type	Number of Isolates	Isolation Source	Year	Country	Reference
*mcr-1*	*E. coli*	ST354	4	Fecal sample	2016	China	[84]
*mcr-1*	*E. coli*	/	5	Fecal sample	2017	China	[85]
*mcr-1*	*E. coli*	ST10	1	Fecal sample	2017	China	[86]
*mcr-1*	*E. coli*	/	45	Nasal and rectal swabs	2017	China	[87]
	*K. pneumoniae*	/	2				
*mcr-1*	*E. coli*	ST93	4	Urine, nasal secretion, fecal sample, diarrhea	2018	China	[35]
		ST1011	1				
		ST3285	1				
		New ST	1				
*mcr-1*	*E. coli*	ST770	1	Urinary tract infection	2019	Argentina	[88]
*mcr-1*	*E. coli*	ST6316	1	Uterus	2019	China	[89]
		ST405	1				
		ST46	1	Urine			
*mcr-1*	*E. coli*	ST162	1	Clinical sample	2019	China	[90]
*mcr-1*	*E. coli*	/	1	Fecal sample	2019	Ecuador	[91]
*mcr-1*	*K. pneumoniae*	ST307	2	Urine, pyometra	2019	Taiwan	[92]
	*E. cloacae*	ST1005	2	Urine			
*mcr-1*	*E. coli*	/	1	Urine	2020	Brazil	[93]
	*Klebsiella* sp.	/	1	Abdominal seroma			
	*Enterobacter.* sp.	/	1	Nasal secretion			
*mcr-1/mcr-3.7*	*E. coli*	ST132	1	Fecal sample	2020	China	[94]
*mcr-1*	*E. coli*	ST1630	1	Rectal swabs	2020	Ecuador	[95]
		ST2170	1				
*mcr-1*	*E. coli*	ST162	1	Fecal sample	2020	Ecuador	[96]
		ST1196	1				
		ST744	1				
*mcr-1*	*E. coli*	ST162	1	Diarrhea	2020	South Korea	[97]
*mcr-1*	*K. pneumoniae*	/	149	Fecal sample	2021	China	[100]
*mcr-1*	*E. coli*	ST648	3	Rectal swabs	2021	China	[98]
		ST23	1				
		ST162	1				
		ST2722	1				
		ST2325	1				
		ST1638	1				
		ST8680	1				
		ST7331	1				
*mcr-1*	*K. pneumoniae*	ST656	1	Urine	2021	China	[99]
*mcr-2*	*K. pneumoniae*	/	11	Fecal sample	2021	China	[100]
*mcr-3*	*K. pneumoniae*	/	15	Fecal sample	2021	China	[100]
*mcr-3*	*E. coli*	ST10	1	Clinical sample	2021	Taiwan	[101]
*mcr-4*	*K. pneumoniae*	/	6	Fecal sample	2021	China	[100]
*mcr-5*	*K. pneumoniae*	/	18	Fecal sample	2021	China	[100]
*mcr-8*	*K. pneumoniae*	ST3410	1	Nasal swabs	2021	China	[99]
*mcr-9*	*K. pneumoniae*	/	5	Fecal sample	2021	China	[100]
*mcr-9*	*E. hormaechei*	ST493	2	Clinical sample	2021	Egypt	[102]
*mcr-9*	*E. coli*	ST372	1	Clinical sample	2021	United Kingdom	[103]
*mcr-10*	*K. pneumoniae*	/	4	Fecal sample	2021	China	[100]
*mcr-10*	*E. roggenkampii*	/	1	Pus	2021	Japan	[104]
*mcr_1*	*E. coli*		10	Fecal sample	2020	France	[105]

**Table 3 pathogens-11-00698-t003:** The *mcr*-positive isolates detected in cats.

*mcr* Genes	Species	ST	Number of Isolates	Source	Year	Country	Reference
*mcr-1*	*E. coli*	ST93	1	Fecal sample	2016	China	[84]
		New ST	1				
*mcr-1*	*E. coli*	/	1	Nasal and rectal swabs	2017	China	[87]
*mcr-1*	*E. coli*	ST93	1	Diarrhea	2018	China	[35]
*mcr-1*	*K. pneumoniae*	ST307	1	Urinary tract infection	2021	Brazil	[106]
*mcr-9*	*E. hormaechei*	ST493	1	Clinical samples	2021	Egypt	[102]
		ST182	2				
*mcr-9*	*E. asburiae*	/	1	Nasal swab	2021	Japan	[107]
*mcr-1*	*E. coli*		4	Fecal sample	2020	France	[105]
*mcr-1*	*Rahnella aquatili*		1	Fecal sample	2020	France	[105]

## Data Availability

Not applicable.

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
