# Peer review of "Mobile Colistin Resistance (mcr) Genes in Cats and Dogs and Their Zoonotic Transmission Risks"

_pathogens, 2022, doi:10.3390/pathogens11060698_

Round 1

Reviewer 1 Report

This is a systematic review about colistin resistance through mcr genes in pets, specially cats and dogs and also highlights the risk of their zoonotic transmission.

The abstract is appropriate, allowing the reader to know what has been done, the review’s aim and the main results and conclusions.

In the introduction authors give the necessary information and the aim of the study is clearly defined.

In our opinion point 6. “Material and methods” must come immediately after the introduction.

The number of papers in PubMed and Web of Science referred in figure 1 are different from the number of papers in these databases referred in table 1. Please verify.

In table 1 title, authors should include in the keywords “pets”, since in point 6.1 is one of keywords referred.

In points 4.1 and 4.2 authors describe all the mcr genes detected in dogs and cats respectively, except the ones from reference 102, that are only indicated in tables 2 and 3.  Is there any reason for this?

Conclusions are in line with the objectives defined for the work.

Minor revisions:

Along the manuscript, the names of species and genes must be in italics

 Line 74 – “research” instead of “resaerch”

Line 74 - We suggest, “The bibliography research resulted in a total of 1231 articles…” instead of “The bibliography research resultes to a total of 1231 articles…”

Line 171 – “Antimicrobial use (AMU) and antimicrobial resistance (AMR) are the most effective issue for human health.” Authors should reformulate or delete this sentence.

Line 195 – Please check this sentence “while 19 (18.7%) persons expressed the same concern.

Line 252 – “Wang et al. (2021) have…” instead of “Wang et al. have…”

Line 273-274 – “The other mcr gene detected in cats was mcr-9 in Enterobacter hormaechei ST493…” instead of “The other mcr gene detected in cats in mcr-9 was detected in Enterobacter hormaechei ST493…”

Line 314 – “Zhang et al. (2016) suggested…” instead of “Zhang et al. suggested…”

Line 394-395 – “Reducing the use of all antibiotics particularly colistin, would be detrimental to human health”. In our opinion this sentence needs contextualization and explanation. This way authors may be given contradictory information.

Line 401- “…strict hygiene conditions must be observed to ensure uncontaminated…” instead of “…strict hygiene conditions must be observed ensure uncontaminated…”

Reviewer 2 Report

  1. The use of polymyxins in animals(pets) and humans should be mentioned in introduction.
  2. Line 51, The PhoP/PhoQ two component system or others systems should be described.
  3. Line 59, a reference should be added.
  4. Line 60, are the “colistin resistance genes”mcr and its variants or others?
  5. Line 66-67, How do humans and animals exchange resistance genes?
  6. Line 80, why were the n=104 articles excluded.
  7. 1, Inconsistent format
  8. For Chromosomic colistin resistance mechnisms, s simple diagram would be much more intuitive, wheremgrB or others genes should also be mentioned.
  9. In the articles, all format of genes and genus should be italicized, such as line 127,130, 132-133, etc. Questions like this are all over the article, a very unprofessional way of writing.
  10. Line 137, a reference should be provided.
  11. In 3.1 ine 145-169, the use of colistin as feed additives wasn’t mentioned.
  12. In the articles, two 3. (3. Antimicrobial uses and3. Colistin resistance mechanisms )titles were involved
  13. Line 173, “is”should be revised as “are”.
  14. In the articles, the author should pay special attention to the tense, which is very confusing.
  15. Line 203, remove “the”before Netherlands.
  16. Line 218, what is the “plasmid-mediated colistin resistance”.
  17. Line 225, “.”should be added after [97].

Round 2

Reviewer 2 Report

The author has made a lot of revisions, and it is suggested to polish the language, the paper will be more perfect.

Author Response

As suggested, the manuscript has been submitted to a professional English proofreading service to polish properly the language. The text has been carefully checked for correct use of grammar and common technical terms.